# On Entropy Regularized Path Integral Control for Trajectory Optimization

**DOI:** 10.3390/e22101120

**Published:** 2020-10-03

**Authors:** Tom Lefebvre, Guillaume Crevecoeur

**Affiliations:** 1Department of Electromechanical, Systems and Metal Engineering, Ghent University, 9000 Ghent, Belgium; guillaume.crevecoeur@ugent.be; 2EEDT Decision & Control, Flanders Make, 3001 Heverlee, Belgium

**Keywords:** entropic inference, entropy regularization, stochastic search methods, path integral control

## Abstract

In this article, we present a generalized view on Path Integral Control (PIC) methods. PIC refers to a particular class of policy search methods that are closely tied to the setting of Linearly Solvable Optimal Control (LSOC), a restricted subclass of nonlinear Stochastic Optimal Control (SOC) problems. This class is unique in the sense that it can be solved explicitly yielding a formal optimal state trajectory distribution. In this contribution, we first review the PIC theory and discuss related algorithms tailored to policy search in general. We are able to identify a generic design strategy that relies on the existence of an optimal state trajectory distribution and finds a parametric policy by minimizing the cross-entropy between the optimal and a state trajectory distribution parametrized by a parametric stochastic policy. Inspired by this observation, we then aim to formulate a SOC problem that shares traits with the LSOC setting yet that covers a less restrictive class of problem formulations. We refer to this SOC problem as Entropy Regularized Trajectory Optimization. The problem is closely related to the Entropy Regularized Stochastic Optimal Control setting which is often addressed lately by the Reinforcement Learning (RL) community. We analyze the theoretical convergence behavior of the theoretical state trajectory distribution sequence and draw connections with stochastic search methods tailored to classic optimization problems. Finally we derive explicit updates and compare the implied Entropy Regularized PIC with earlier work in the context of both PIC and RL for derivative-free trajectory optimization.

## 1. Introduction

Finding controllers for systems that are high dimensional and continuous in space is still one of the most challenging problems faced by the robotic and control community. The goal is to find a feedback policy that stabilizes the system and that possibly also encodes rich and complex dynamic behavior such as locomotion [1]. A feedback policy is a state- and time-dependent function u(t,x) with *u* representing the input applied to the system that is in a state *x* at time *t*. A well-known paradigm to design such policies is Stochastic Optimal Control (SOC) [2]. The policy is determined such that when applied to the system, it is expected to accumulate a minimized cost over a specified finite time horizon. Finding such an optimal feedback policy is an elegant and appealing theoretical idea but meets significant difficulties in practice. Explicit expressions for the optimal policy $u^{*}\left(t,x\right)$ rarely exist and one often has to resort to local solutions instead. Such local solutions can be found by so called trajectory optimization algorithms (In this work we focus on discrete-time system and model-based strategies, in the sense that a simulator is available that closely approximates the actual dynamics) [3,4]. These yield open-loop solutions of the form $u^{*}\left(t\right)=u^{*}\left(t,x^{*}\left(t\right)\right)$ where the signals $u^{*}\left(t\right)$ and $x^{*}\left(t\right)$ are optimal in the sense that they minimize the accumulated cost starting from a given fixed initial state.

In hierarchical control approaches such trajectory optimizers are used to discover rich and complex dynamic behaviors “offline” [5]. The system is then stabilized “online” using a linearized feedback-controller around the optimal trajectory. The locally linear feedback-controller is usually designed such that $u_{\mathrm{fb}}^{*}\left(t,x\right)=k\left(t\right)+K\left(t\right)x\left(t\right)=u^{*}\left(t\right)+K\left(t\right)\left(x\left(t\right)-x^{*}\left(t\right)\right)$ where $K\left(t\right)\approx \nabla_{x}u\left(t,x^{*}\left(t\right)\right)$ and $k\left(t\right)=u^{*}\left(t\right)+K\left(t\right)x^{*}\left(t\right)$. The gradient K(t) is approximated by defining a Linear Quadratic Regulator (LQR) problem around the optimal trajectory using a second order Taylor expansion. In fact, this LQR problem then also provides a correction to $\left\{x^{⋆}\left(t\right),u^{⋆}\left(t\right)\right\}$ were this linearization point may not be optimal. As such, a new trajectory $\left\{u^{⋆}\left(t\right),x^{⋆}\left(t\right)\right\}$ is obtained around which a new LQR can be defined. Iterating this approach will incrementally improve the trajectory. This is exactly how gradient based trajectory optimization algorithms work such as Differential Dynamic Programming (DDP) [3,6] and iterative LQR (iLQR) [7]. It is implied that gradient based trajectory optimizers require a differentiable model and cost function. Furthermore, they are notoriously ill-suited to handle state constraints. A second application is that of Model Predictive Control (MPC). The idea is to compute an optimal trajectory initialized with the current state measurement and apply the optimal open-loop controller every sample instant. This approach provides a “just-in-time” service and circumvents computation of the explicit policy altogether. MPC is feasible provided that the trajectory optimization problem can be solved in a single sample period [8]. This poses hard real-time requirements on the trajectory optimizer that are rarely met in practice, specifically for nonlinear dynamics with state and input constraints.

### 1.1. Stochastic Search Methods

In this article, we are interested in solving the trajectory optimization problem above using a sample based optimization method, or a so-called stochastic search method. Stochastic search methods rely on randomness to probe the optimization space and maintain mechanisms that eventually guide that randomness towards prosperity. Broadly speaking, a stochastic search method maintains a prior population and generates a posterior population based on the prosperity of the individuals.

Evolutionary Strategies (ESs) refer to a particular subclass of stochastic search algorithms tailored to static optimization, that, as opposed to population based algorithms [9], engage a parametric (θ) *search distribution* model, $\pi \left(x|\theta \right):X\mapsto R_{\geq 0}$. Every main generation, *g*, a sample population, $X_{g}=\;\left\{x_{g}^{k}\right\}$, is generated from a search distribution $\pi (x|\theta_{g})$ and the search distribution parameters, $\theta_{g+1}\leftarrow \theta_{g}$, are updated based on the relative success of the individual samples. The objective function value $f(x^{k})$ is used as a discriminator between prosperous and poor behavior of each individual, $x^{k}$ [10]. When iterated this concept spawns a sequence of distributions, $\left\{\pi_{g}\right\}$. The update procedure is devised so that the distribution sequence migrates gradually through the optimization space and concentrates around the optimal solution eventually. Although that limiting the sequence to a parametric distribution family may compromise the inherent expressiveness or elaborateness of the associated search, it also elevates the determination of update rules, from what are basically heuristics, to a more rigorous and theoretical body [11,12]. Well-known members are the Covariance Matrix Adaptation Evolutionary Strategy (CMA-ES) [11,13] and the Natural Evolutionary Strategy (NES) [14]. Most ESs consider a multivariate Gaussian parametric distribution and provide appropriate update procedures for the mean, $\mu_{g}$, and covariance, $\Sigma_{g}$. Examples of application on medium to high scale problems for non-differential design optimization are [15,16,17,18,19,20,21].
$$\pi_{g}\left(x\right)=N\left(\theta_{g}\right)=N\left(x|\mu_{g},\Sigma_{g}\right)$$

In the present work, we aim to address trajectory optimization problems, exploiting stochasticity as a natural means of exploration, in a similar fashion as how ESs address static optimization problem. The trajectory optimization problem is however fundamentally different in the sense that we can no longer probe the optimization space directly. We can only do so by applying stochastic policies to the system, then observe how the system evolves and infer an updated search distribution from these system *rollouts*.

Specifically, we are interested in locally linear Gaussian feedback policies of the following form, that we can apply to the system to artificiality inject the required stochasticity. Clearly this policy resembles the locally linear feedback trajectory as was describe in the previous paragraph and the idea can in that sense be understood as a sample based implementation of the iLQR or DDP algorithm.
$$\pi_{g,t}\left(u|x\right)=N\left(u|x;\theta_{g,t}\right)=N\left(u|k_{g,t}+K_{g,t}x,\Sigma_{g,t}\right)$$

The covariance determines the explorativeness of the policy and therefore the diverseness of the samples. Our goal is to actively shape it to stimulate exploration.

### 1.2. Path Integral Control

In the previous decade, a novel class of stochastic search algorithms was discovered that partially answer our question. This class is known as Path Integral Control (PIC). PIC is closely related to the Linearly Solvable Optimal Control (LSOC) prolem. LSOC is a restrictive subset of the Stochastic Optimal Control (SOC) framework (see Section 2) and is characterized by a number of remarkable properties (see Section 3.1) [22,23,24]. It was already pointed out that PIC and ES exhibit structural similarities [25,26]. This lead to the development of a PIC-CMA algorithm. This method deviates from the theory of LSOC and adapts the policy covariance in analogy with CMA-ES. This modification improved the convergence properties significantly yet ignores the underlying theory of LSOC. Other attempted generalizations include [27,28].

There has been keen interest in such algorithms as stochastic search algorithms may exhibit several advantages over gradient based algorithms [25,26,28]. So far it has been used in guided policy search to generate a set of prior optimal trajectories that were then used to fit a global policy [29] and is one of the two algorithms promoted by the *Lyceum* robot learning environment [30]. The use of PIC algorithms for real-time control applications was only recently considered as their execution is similar to Monte Carlo (MC) algorithms and were therefore thought to be too time critical to perform in real-time. However, with the rise of affordable GPUs and the ease of parallelization of MC based methods, it may become feasible in the near future to iterate dynamic stochastic search algorithms in real-time [31,32]. Nevertheless, practitioners of such methods have raised issues concerning the update of the covariance matrix [25,29,31,32]. It seems a mechanism is inherent to the existing framework that makes the covariance matrix vanish, compromising exploration. In other words, the search distribution collapses prematurely. Many authors suggested that the issue is limitedly understood and that it seems unlikely that it can be resolved with the theory at hand.

Furthermore, it is clear that ESs and PIC methods are closely related and that therefore PIC method may benefit from the rich body of work concerning ESs. However, as PIC are derived solely from the theory of LSOC, we argue that they are only limitedly understood and their similarity has been circumstantial. Recently, Williams et al. provide a novel derivation of the PI^2^ method from an information-theoretic background [32]. Other authors have explored the relation between LSOC and information-theory [32,33,34], however these studies aimed for a physical connection and understanding.

In this paper, we venture on a different strategy and aim to identify a generalized set of PIC methods. We aim to describe an overarching optimization principle that leads us to derive to ESs in the context of static optimization and PIC methods in the context of dynamic optimization. Therefore, we approach the problem from an algorithmic point of view, rather than searching for a deeper physical interpretation or understanding. To establish the overarching framework, we identify the principle of entropic inference as a suitable setting to synthesize stochastic search algorithms and derive an entropic optimization framework from it. This will also allow us to derive a generalized set of PIC methods which are no longer limited to the LSOC setting and therefore do not inherit any of its inherent limitations. Furthermore, the mutual theoretical background paves way for a knowledge transfer from ES to PIC. Finally, this viewpoint provides us with a unique opportunity to relate PIC to existing Entropy Regularization paradigms in Reinforcement Learning [35,36,37,38].

### 1.3. Contributions

As our efforts span a rather extensive body of earlier work and aim to provide a holistic view of related subjects, we feel inclined to list our original contributions.

We provide a comprehensive overview of existing Path Integral Control tailored to policy search. Here we point out a mutual underlying design principle which allows for a formal comparison and classification.We propose an original and intuitive argument for the introduction of entropy regularization terms in the context of optimization that is based on the principle of entropic inference.We untie the derivation of Path Integral Control methods from its historical roots in Linearly Solvable Optimal Control and illustrate that a similar set of algorithms can be derived directly from the more timely framework of Entropy Regularized Optimal Control. Therefore, we introduce the framework of Entropy Regularized Trajectory Optimization and derive the Entropy Regularized Path Integral Control (ERPIC) method. We consider this to be our primary contribution. Furthermore, this work elevates the structural similarity between Evolutionary Strategies, such as the CMA-ES [13] and NES [14], and PIC methods originally pointed out and exploited in [25], to a formal equivalence.We give a formal comparison of preceding PIC methods and ERPIC tailored to derivative-free trajectory optimization with control affine dynamics and locally linear Gaussian policies.

## 2. Preliminaries and Notation

Let us first introduce a number of relevant concepts and establish notation.

### 2.1. General Notation

We denote the set of continuous probability density functions over any closed set $X\subseteq R^{n}$ as $P\left(X\right)=\{p:X\mapsto R_{\geq 0}|\int_{X}\pi \left(x\right)dx=1\}$. We will not always specify the argument as it is implied by the distribution definition. The statement π∝p with π∈P denotes that π is proportional to *p* up to a normalization constant. The expectation of *f* over a probability density function π is denoted as $E_{\pi }\left[f\right]=\int_{X}\pi \left(x\right)f\left(x\right)dx$. We will also need a number of information theoretic measures.

The differential entropy is defined as
$$H\left[\pi \right]=E_{\pi }\left[-\log\pi \right]=-\int \log\pi \left(x\right)\pi \left(x\right)dx$$

This is widely interpreted as a metric of uncertainty. However, more precisely it is to say that it is a measure for the amount of information left to be specified about some epistemological uncertain variable *X* for which we have represented our belief with a probability density function π [39].

The Kullback–Leibler divergence or relative entropy between two probability density functions π and ρ is defined as
$$D\left[\pi \|\rho \right]=E_{\pi }\left[\log\frac{\pi }{\rho }\right]=\int_{X}\pi \left(x\right)\log\frac{\pi (x)}{\rho (x)}dx\geq 0$$

The relative entropy is always positive but not symmetric $D\left[\pi \|\rho \right]\neq D\left[\rho \|\pi \right]$. An interpretation is given in Section 4.

### 2.2. Dynamic System Models

Further, we will consider controlled stationary deterministic or stochastic discrete-time systems. We use x∈X to represent the system state and u∈U to denote the input or control effort. Deterministic systems are modeled with a state-space function $x_{t+1}=f\left(x_{t},u_{t}\right)$, f:X×U↦X. Stochastic systems are modeled by a state transition distribution function $x_{t+1}∼p\left(x_{t+1}|x_{t},u_{t}\right)$, $p:\;X\times X\;\times \;U\mapsto R_{\geq 0}$. We also define two classes of non-stationary feedback policies. Deterministic policies are denoted as $u_{t}=u\left(t,x_{t}\right)$, u:Z×X↦U, and stochastic policies are denoted as $u_{t}∼\pi \left(u_{t}|t,x_{t}\right)$, $\pi :Z\times U\times X\mapsto R_{\geq 0}$. Note that any deterministic dynamic system can also be represented as a stochastic system using the Dirac delta, $p\left(x^{\prime }|x,u\right)=\delta \left(x^{\prime }-f\left(x,u\right)\right)$. Analogously, any deterministic policy can also be represented as a stochastic policy, $\pi \left(u_{t}|t,x_{t}\right)=\delta \left(u-u\left(t,x_{t}\right)\right)$.

A realization of a system with policy *u* or π over a horizon *T* results into a trajectory $\tau =\left\{x_{0},u_{0},x_{1},u_{1},\cdots ,x_{T-1},u_{T-1},x_{T}\right\}\in T\subset S_{1:T}\times A_{0:T-1}$ or a state trajectory $\tau_{x}=\left\{x_{0},x_{1},\cdots ,x_{T-1},x_{T}\right\}\in T_{X}\subset S_{1:T}$. In any case, we will assume that trajectories τ∈T agree with the underlying system dynamics and is induced by a stochastic (or deterministic) policy.

Considering that we work within a stochastic setting, we can associate a probability to each trajectory. The trajectory distribution or path distribution induced by a policy π is denoted as p(τ|π). The state trajectory distribution induced by a policy π is denoted as $p_{\pi }\left(\tau_{x}\right)$. These distributions can be factorized in a product of transition probabilities.
$$\begin{array}{cc} p(\tau |\pi ) & =\prod_{t=0}^{T-1}p\left(x_{t+1}|x_{t},u_{t}\right)\pi \left(u_{t}|t,x_{t}\right)\\ p_{\pi }\left(\tau_{x}\right) & =\prod_{t=0}^{T-1}p_{\pi }\left(x_{t+1}|t,x_{t}\right) \end{array}$$

Here, we defined the controlled state transition distribution. Note that the control effort is now entirely explicit.
$$p_{\pi }\left(x_{t+1}|t,x_{t}\right)=\int p\left(x_{t+1}|x_{t},u_{t}\right)\pi \left(u_{t}|t,x_{t}\right)du_{t}$$

This definition also implies two state trajectory distributions of particular interest. The distribution $p_{u}$ induced by a deterministic policy *u* and the uncontrolled or free trajectory distribution which we denote as $p_{0}$.

We can associate a cost to a trajectory τ∈T. In a general setting, we consider the cost R:T↦R that accumulates at a rate r:S×A↦R.
$$R\left(\tau \right)=r_{T}\left(x_{T}\right)+\sum_{t=0}^{T-1}r_{t}\left(x_{t},u_{t}\right)$$

Analogously, we can associate a cost $C:T_{X}\mapsto R$ to state trajectories $\tau_{x}\in T_{X}$. Here, the control efforts is accounted for implicitly through the accumulation rate c:X×X↦R that penalizes transitions $x\to x^{\prime }$ (Note that we will often change between the use of time subscripts and the prime notation to indicate an increment in the time dimension)
$$C\left(\tau_{x}\right)=c_{T}\left(x_{T}\right)+\sum_{t=0}^{T-1}c_{t}\left(x_{t},x_{t+1}\right)$$

If there exists an inverse dynamic function $f^{-1}$ so that $u=f^{-1}\left(x,x^{\prime }\right)$ when $x^{\prime }=f\left(x,u\right)$, *C* and *R* are equivalent in the sense that $r_{t}\left(x,f^{-1}\left(x,x^{\prime }\right)\right)=c_{t}\left(x,x^{\prime }\right)$ and $r_{T}\left(x\right)=c_{T}\left(x\right)$. We emphasize here that the existence of such an inverse model is a mere formal assumption. In the practical setting to be presented, we will never actually have to invert the system dynamics given that we will have access to *u*.

Finally, we are also interested in the exotic cost function $L:T_{X}\mapsto R$. The total cost can be decomposed as the superposition of a cost $\tilde{L}$ that accumulates with a solely state dependent rate l:X↦R and a term that accounts for the control effort. Specifically we are interested in the situation where the control efforts are penalized indirectly by introducing the logarithm of the ratio between the controlled and free state transition probabilities. This specific choice will have a remarkable technical consequence later on. Further, note that *L* is therefore of the same type as *C* and operates on $T_{X}$. It follows that if the transition $x\to x^{\prime }$ is likelier when the system is controlled, i.e., $p_{u}\left(x^{\prime }|x\right)>p\left(x^{\prime }|x\right)$ the ratio is larger than 1 and so its logarithm is positive, effectively inducing a cost. Alternatively, when the transition becomes less probable, this induces a negative penalty, again inducing a cost. As a result it will depend on $\tilde{L}$ whether the use of $p_{u}$ over $p_{0}$ is beneficial or not.
$$L\left(\tau_{x}\right)=\underset{\tilde{L}(\tau_{x})}{\underset{︸}{l_{T}\left(x_{T}\right)+\sum_{t=0}^{T-1}l_{t}\left(x_{t}\right)}}+\sum_{t=0}^{T-1}\log\frac{p_{u}\left(x_{t+1}|t,x_{t}\right)}{p_{0}\left(x_{t+1}|t,x_{t}\right)}$$

### 2.3. Stochastic Optimal Control

We are interested in finding policies that minimize the expected induced cost. Hence we address the Stochastic Optimal Control (SOC) problem. Such policies are known to be stabilizing and are capable of encoding very rich and complex dynamic behavior even for considerably simple cost formulations.
(1)$$u^{*}=\arg\min_{u}E_{p(\tau |u)}\left[R\left(\tau \right)\right]$$

Solving (Equation 1) poses a so-called dynamic optimization problem. The prefix dynamic emphasizes that the optimization variables are constrained by a causal structure which allows to break the problem apart into several subproblems that can be solved recursively. This problem property is also known as *optimal substructure* which can be exploited by dedicated solution algorithms.

Correspondingly, the problem above can also be represented with the recursive Bellman equation. Here, $V_{t}:X_{t}\mapsto R$ represents the value function or optimal cost-to-go, i.e., the cost that is accumulated if we initialize the system in state *x* at time *t* and control it optimally until the horizon *T*.
$$\begin{array}{c} V\left(t,x\right)=\min_{\pi }\int \pi \left(u|t,x\right)\left(r_{t}\left(x,u\right)+\int V\left(t+1,x^{\prime }\right)p\left(x^{\prime }|t,x,u\right)dx^{\prime }\right)du \end{array}$$

We emphasize that we presume the policy to be stochastic. However, unless there is the incentive to maintain a stochastic policy for the purpose of exploration, the expression minimizes for a deterministic policy so that the expectation over the policy can be omitted [40]. This redundant form is however appealing for later comparison. Further, note that if also $p\left(x^{\prime }|x,u\right)=\delta \left(x^{\prime }-f\left(x,u\right)\right)$, this problem reduces to deterministic optimal control.
(2)$$\begin{array}{c} V\left(t,x\right)=\min_{u}r_{t}\left(x,u\right)+\int V\left(t+1,x^{\prime }\right)p\left(x^{\prime }|x,u\right)dx^{\prime } \end{array}$$

Throughout this manuscript we will further assume that the initial state is fixed and known exactly, i.e., $x\left(0\right)=x_{0}$. We could stress this formally with our notation and exemplify that any function associated to problem (Equation 2), be it u(t,x)π(u|t,x) or V(t,x) or any of the implied functions $p_{\pi }\left(x^{\prime }|t,x\right)$ and $p_{\pi }\left(\tau_{x}\right)$ are conditional on $x_{0}$. In order not to overload the notational burden we simply assume this to be true throughout the paper. This assumption will allow us to concentrate on local solutions or so called trajectory optimizers.

### 2.4. Local Parametric Policies

The algorithms we present here do not aim to retrieve an exact solution to the problem above, but instead operate on a restricted class of parametrized policies π(u|t,x,θ) where θ refers to the policy parameter. Considering that we are interested in trajectory optimizers, we restrict our focus to locally linear Gaussian policies of the form $\pi \left(u|t,x,\theta \right)=N\left(u|u_{t}+K_{t}x,\Sigma_{t}\right)$, so that $\theta ={\left\{\theta_{t}\right\}}_{t=0}^{T}$ with $\theta_{t}=\left\{u_{t},K_{t},\Sigma_{t}\right\}$. Here $K_{t}$ represented a feedback gain matrix and $\Sigma_{t}$ the covariance matrix which controls the stochasticity and therewith the explorative incentive of the policy. This parametrization allows to approximate the exact solutions in the neighborhood of an optimal open-loop trajectory initialized at for example $x\left(0\right)=x_{0}$.

## 3. Path Integral Control

In this section we give a brief introduction to the theory of LSOC and give an overview of existing PIC methods rooting directly from it. In almost all preceding contributions (apart from those following [22]), LSOC is addressed in a continuous time setting and a discretization is only performed afterwards in order to derive practical methods. Here we will directly address the LSOC problem in a discrete time setting as it better suits our requirements and as such also avoid the tedious technicalities involved with the discretization.

Second, we give a comprehensive overview of the class of PIC methods. As far as we are aware of, this is the first time such a formal comparison is made, revealing an overarching design principle shared by all algorithmic formulations. The identification of this design principle makes it easier to pinpoint the core prerequisite of any PIC method and will allow us to introduce a generalized set of PIC methods in Section 4.3.

### 3.1. Discrete Time Linearly Solvable Optimal Control

Formally, the discrete time LSOC framework refers to the following SOC problem,
(3)$$\begin{array}{cc} V(t,x) & =\min_{u}l_{t}\left(x\right)+\int p\left(x^{\prime }|t,x,u\right)\left(\log\frac{p_{u}\left(x^{\prime }|t,x\right)}{p_{0}\left(x^{\prime }|t,x\right)}+V\left(t+1,x^{\prime }\right)\right)dx^{\prime }\\ \; & =\min_{u}l_{t}\left(x\right)+D\left[p_{u}\|p_{0}\right]+E_{p_{u}}\left[V\left(t+1,x^{\prime }\right)\right] \end{array}$$

We emphasize that the control effort is penalized implicitly through the Kullback–Leibler divergence between the controlled and free state transition probabilities. For a formal motivation for this particular penalty formulation, we refer the reader to the work in [22,33] and recall the intuitive justification that was already given in Section 2. Here, we are mostly interested in the technical implications that are associated with this problem formulation.

Problem (Equation 3) implies the existence of an optimal policy $u_{\mathrm{LSOC}}^{*}\left(t,x\right)$
$$u_{\mathrm{LSOC}}^{*}\left(t,x\right)=\arg\min_{u}l_{t}\left(x\right)+D\left[p_{u}\|p_{0}\right]+E_{p_{u}}\left[V\left(t+1,x^{\prime }\right)\right]$$

In general the problem can not be solved exactly for $u_{\mathrm{LSOC}}^{*}\left(t,x\right)$. However, the following theorem establishes a relation between the optimal policy $u_{\mathrm{LSOC}}^{*}\left(t,x\right)$, the optimal state transition distribution $p_{u_{\mathrm{LSOC}}^{*}}\left(x^{\prime }|t,x\right)$ and the optimal state trajectory distribution $p_{u_{\mathrm{LSOC}}^{*}}\left(\tau_{x}\right)$, and summarizes the most profound results in the context of LSOC.

**Theorem** **1.**
*With V(t,x) defined as in (Equation 3), the following problems are equivalent,*
(4)$$\begin{array}{cc} u_{\mathrm{LSOC}}^{*}\left(t,x\right)= & \arg\min_{u}l_{t}\left(x\right)+D\left[p_{u}\|p_{0}\right]+E_{p_{u}}\left[V\left(t+1,x^{\prime }\right)\right] \end{array}$$
(5)$$\begin{array}{cc} p_{u_{\mathrm{LSOC}}^{*}}\left(x^{\prime }|t,x\right)= & \arg\min_{p_{u}\in P}l_{t}\left(x\right)+D\left[p_{u}\|p_{0}\right]+E_{p_{u}}\left[V\left(t+1,x^{\prime }\right)\right] \end{array}$$
(6)$$\begin{array}{cc} p_{u_{\mathrm{LSOC}}^{*}}\left(\tau_{x}\right)= & \arg\min_{p_{u}\in P}E_{p_{u}}\left[L\right]=E_{p_{u}}\left[\tilde{L}\right]+D\left[p_{u}\|p_{0}\right] \end{array}$$

*The latter can be solved explicitly*
$$\begin{array}{cc} p_{u_{\mathrm{LSOC}}^{*}}\left(x^{\prime }|t,x\right) & \propto p_{0}\left(x^{\prime }|t,x\right)e^{-V(t+1,x^{\prime })}\\ p_{u_{\mathrm{LSOC}}^{*}}\left(\tau_{x}\right) & \propto p_{0}\left(\tau_{x}\right)e^{-\tilde{L}\left(\tau_{x}\right)} \end{array}$$

*where V(t,x) is governed by the recursion*
$$V\left(t,x\right)=l_{t}\left(x\right)+\log\int p_{0}\left(x^{\prime }|t,x\right)e^{-V(t+1,x^{\prime })}dx^{\prime }$$

*and where $u_{\mathrm{LSOC}}^{*}\left(t,x\right)$ and $p_{u_{\mathrm{LSOC}}^{*}}\left(x^{\prime }|t,x\right)$ are related as*
$$p_{u_{\mathrm{LSOC}}^{*}}\left(x^{\prime }|t,x\right)=p\left(x^{\prime }|x,u_{\mathrm{LSOC}}^{*}\left(t,x\right)\right)$$


The proof relies on the calculus of variations. These peculiar results follow directly from the fact that the problem does no longer depend on *u* explicitly. For details we refer the reader to earlier references.

Most remarkable is that the equivalence of problems (Equation 4) and (Equation 5) implies that we can lift the optimization problem from the control space to the state transition distribution space. Second, although we can not solve to $u_{\mathrm{LSOC}}^{*}\left(t,x\right)$ directly, we can solve for the induced optimal state transition distribution, $p_{u_{\mathrm{LSOC}}^{*}}\left(x^{\prime }|t,x\right)$. Note that therefore we should still identify V(t,x). The equivalence of (Equation 6) and (Equation 6) follows from the recursive definition of V(t,x) that can be evaluated exactly, and thus implies the existence of an explicit optimal state trajectory distribution, $p_{u_{\mathrm{LSOC}}^{*}}\left(\tau_{x}\right)$ also. This is a unique trait of the LSOC setting and lies at the very root of every PIC method. The reason why there exists an explicit solution is a direct consequence from the control effort penalization which makes it possible to lift the optimization problem. As an unfortunate side effect, the system uncertainty and the control penalization are now somehow entangled. This we identify as a fundamental limitation of the LSOC setting and related methods.

A final remark can be made with respect to control affine system dynamics, $x^{\prime }=a\left(x\right)+B\left(x\right)u$, disturbed by unbiased Gaussian noise ξ∼N(ξ|0,Σ) in the control subspace. In this case, it is possible to derive an explicit expression for the optimal control policy $u_{\mathrm{LSOC}}^{*}\left(t,x\right)$. The controlled transition distribution is then given by $p\left(x^{\prime }|x,u\right)=N\left(x^{\prime }|a\left(x\right)+B\left(x\right)u,B\left(x\right)\Sigma B{\left(x\right)}^{⊤}\right)$ and the control policy $u_{\mathrm{LSOC}}^{*}\left(t,x\right)$ can be extracted by comparing the expected value of $x^{\prime }$ based on either distribution $p(x^{\prime }|x,u_{\mathrm{LSOC}}^{*}\left(t,x\right))$ or $p_{u_{\mathrm{LSOC}}^{*}}\left(x^{\prime }|t,x\right)$.

It is easily verified that
(7)$$\begin{array}{cc} u_{\mathrm{LSOC}}^{*}\left(t,x\right) & =\frac{E_{p_{0}\left(x^{\prime }|t,x\right)}\left[\xi \;\cdot \;e^{-V(t+1,x^{\prime })}\right]}{E_{p_{0}\left(x^{\prime }|t,x\right)}\left[e^{-V(t+1,x^{\prime })}\right]}=\frac{E_{p_{0}\left(\tau_{x}|t,x\right)}\left[\xi \;\cdot \;e^{-\tilde{L}\left(\tau_{x}|t+1,x^{\prime }\right)}\right]}{E_{p_{0}\left(x^{\prime }|t,x\right)}\left[e^{-\tilde{L}\left(\tau_{x}|t+1,x^{\prime }\right)}\right]} \end{array}$$

This is the discrete time version of the property given in for example [41] Theorem 1, and clearly illustrates the *path integral* terminology. Moreover, it follows that the optimal policy can be estimated with Monte Carlo sampling from the free system dynamics making it appealing to use directly in an MPC setting [31,32,42]. However, as noted in the introduction we are mostly interested in its applications as a trajectory optimization method.

In the following section, we detail a number of policy search methods that effectively exploit this peculiar setting. All of them would rely on the stochastic system dynamics to explore the solution space.

### 3.2. Path Integral Control (PIC) Methods

The unique traits of the LSOC framework have been exploited to derive a class of so called PIC methods. The interested reader is referred to earlier references [26,27,28,30,32,41,43,44,45,46,47]. An overview of applications was already given in the introduction. Originally, the optimal policy was estimated directly from (Equation 7) using Monte Carlo samples generated with the free system, i.e., $p_{0}$. Other methods evolved beyond that idea but the basic principle remained the same. In any case the goal is to find the optimal policy $u_{\mathrm{LSOC}}^{*}\left(t,x\right)$ relying on the inherent stochasticity of the system. To be able to give a concise overview, we distilled a generic design principle that allows to derive different PIC methods. Note that the authors have identified this principle and that the derivations included in the references may not explicitly state this concept.

The principle is based on two distributions. We assume the existence of a formal but explicit goal state trajectory distribution, say $p_{u^{⋆}}$, and, use a parametrized path distribution, $p_{u(\theta )}$, induced by some parametric policy, u(θ).

It is then possible to infer the optimal policy by projecting the parametrized distribution $p_{u(\theta )}$ onto the optimal distribution $p_{u^{⋆}}$, and, determine the corresponding optimal policy parameter by minimizing the projection distance. Based on the cross-entropy argument originally made by [41], the Kullback–Leibler divergence is used as a projection operator.

The following optimization problem is addressed.
$$\begin{array}{cc} \min_{\theta }D\left[p_{u^{⋆}}\|p_{u(\theta )}\right] & =-E_{p_{u}^{⋆}}\left[\log p_{u(\theta )}\right]+\mathrm{constant}\\ \; & =-J(\theta |p_{u^{⋆}})+\mathrm{constant} \end{array}$$

Provided that we can sample from the distribution $p_{u^{⋆}}$, this expectation can be approximated using a Monte Carlo estimator. Assuming that we can evaluate but not sample from $p_{u^{⋆}}$, but dispose of a distribution $p_{u_{g}}$ that we can both evaluate and sample from, another estimate can be obtained by the idea of importance sampling. Such a distribution is easily constructed by applying some policy $u_{g}$ to the system. Particularly interesting is the setting where $u_{g}$ is somehow informed about the desired distribution $p_{u^{⋆}}$. We denote this objective with subscript *g* to emphasize the use of the guiding policy $u_{g}$.
$$J_{g}\left(\theta |p_{u^{⋆}}\right)=E_{p_{u_{g}}}\left[\frac{p_{u^{⋆}}}{p_{u_{g}}}\;\cdot \;\log p_{u(\theta )}\right]$$

We introduce the Monte Carlo estimator ${\hat{J}}_{g}\left(\theta |p_{u^{⋆}}\right)$ for the formal objective defined above. As a guiding policy we substitute $u(\theta_{g})$ for $u_{g}$
(8)$$J_{g}\left(\theta |p_{u^{⋆}}\right)\approx {\hat{J}}_{g}\left(\theta |p_{u^{⋆}}\right)=\frac{1}{M}\sum_{k=1}^{M}\frac{p_{u^{⋆}}\left(\tau_{x}^{k}\right)}{p_{u(\theta_{g})}\left(\tau_{x}^{k}\right)}\;\cdot \;\log p_{u(\theta )}\left(\tau_{x}^{k}\right)$$

This framework admits to define an iterative strategy where $\theta_{g+1}$ is found by optimizing the objective ${\hat{J}}_{g}\left(\theta |p_{u^{⋆}}\right)$. As the existence of an explicit path distribution was so far a unique trait of the LSOC setting, the derivation of PIC methods has always been tied to the framework.

Depending on the optimal distribution substituted for $p_{u^{⋆}}$, the parametrization of the policy u(θ) and the strategy used to solve the optimization problem, different PIC methods are obtained that can be useful to tackle different problems. As already stated in the introduction we are interested in its application in the context of trajectory optimization, i.e., finding local optimal control solutions for a fixed given initial state. Therefore, we will be interested in local policy parametrizations.

Regarding the optimization strategy we can make a distinction between two approaches, denoted as exact methods and gradient descent methods.

#### 3.2.1. Exact Methods

For a particular subset of policy representations in combination with a control affine system model, the objective in (Equation 8) can be maximized exactly. If these conditions are met, the exact solution $\theta^{*}$ can be substituted for the next guiding policy parameter $\theta_{g+1}$. We refer to these approaches as exact methods. Such are particularly interesting to find local policies. We refer to Section 5 for details.
$$\theta_{g+1}=\arg\max_{\theta }{\hat{J}}_{g}\left(\theta |p_{u^{⋆}}\right)$$

Regarding the choice for the goal distribution $p_{u^{⋆}}$ we can make an additional distinction between the Sample Efficient Path Integral Control (SEPIC) and Path Integral Relative Entropy Search (PIREPS) method.

##### Sample Efficient Path Integral Control Method

Most convenient is to substitute the desired optimal path distribution $p_{u_{\mathrm{LSOC}}^{*}}$ for $p_{u^{⋆}}$
$$p_{u^{⋆}}=p_{u_{\mathrm{LSOC}}^{*}}$$

This strategy boils down to the one proposed by Williams et al. [30,32,44] and partially with the one proposed in [48] (In [48], the particular estimation of the system dynamics allows for a more general solution in terms of Equation (Equation 7). However this is out of scope here as we are interested in local policy parametrizations). The terminology stems from the fact that the sampling is done using the informed distribution $p_{u_{g}}$ as a guiding post. Furthermore, if the parametric policy is sufficiently expressive to cover the optimal policy $u_{\mathrm{LSOC}}^{*}$, this methods converges to the actual solution of the LSOC framework. The approximate objective is evaluated as
(9)$${\hat{J}}_{g}\left(\theta |p_{u_{\mathrm{LSOC}}^{*}}\right)=\frac{1}{M}\sum_{k=1}^{M}p_{u(\theta_{g})}{\left(\tau_{x}^{k}\right)}^{-1}\;\cdot \;p_{0}\left(\tau_{x}^{k}\right)\;\cdot \;e^{-\tilde{L}\left(\tau_{x}^{k}\right)}\;\cdot \;\log p_{u(\theta )}\left(\tau_{x}^{k}\right)$$

SEPIC aims to solve for $p_{u_{\mathrm{LSOC}}^{*}}$ directly. The problem is that if the original guiding policy is too far removed from the optimal policy, the method will not converge. Because the interesting regions in the solution space are simply not sampled and the optimal policy is never discovered.

##### Path Integral Relative Entropy Policy Search

To remedy this issue [47] proposed PIREPS.

Retrospectively, this is an information-theoretic trust-region strategy which introduces a regularization term penalizing the Kullback–Leibler divergence between the new and old path distribution to promote more conservative policy updates. In fact, this is a form of entropy regularization which we will come back to in Section 4. This idea was introduced independently from [47] in [29] to generate local policies in the context of guided policy search.
$$p_{u_{g+1}}=\arg\min_{p_{u}\in P}D\left[p_{u}\|p_{u_{\mathrm{LSOC}}^{*}}\right]+\lambda D\left[p_{u}\|p_{u_{g}}\right]$$

The approach generates a sequence of intermediate pseudo-optimal path probabilities that aim to improve the convergence properties. This problem can be solved explicitly and the intermediate probabilities are substituted for $p_{u^{⋆}}$ consequently. The idea is that the distribution $p_{u_{g+1}}$ will be closer to the solution space sampled by $p_{u(\theta_{g})}$ and that as a direct result the policy updates will be more robust. Note that for g→∞ the sequence of pseudo-optimal path probabilities collapses on $p_{u_{\mathrm{LSOC}}^{*}}$.
$$p_{u^{⋆}}=p_{u_{g+1}}\propto p_{u_{g}}^{\frac{\lambda }{1+\lambda }}\;\cdot \;p_{0}^{\frac{1}{1+\lambda }}\;\cdot \;e^{-\frac{1}{1+\lambda }\tilde{L}}=p_{u_{0}}^{{\left(\frac{\lambda }{1+\lambda }\right)}^{g+1}}\;\cdot \;p_{u_{\mathrm{LSOC}}^{*}}^{1-{\left(\frac{\lambda }{1+\lambda }\right)}^{g+1}}$$

The approximate objective is evaluated as
(10)$${\hat{J}}_{g}\left(\theta |p_{u_{g+1}}\right)=\frac{1}{M}\sum_{k=1}^{M}p_{u(\theta_{g})}^{\frac{-1}{1+\lambda }}\left(\tau_{x}^{k}\right)\;\cdot \;p_{0}^{\frac{1}{1+\lambda }}\left(\tau_{x}^{k}\right)\;\cdot \;e^{-\frac{1}{1+\lambda }\tilde{L}\left(\tau_{x}^{k}\right)}\;\cdot \;\log p_{u(\theta )}\left(\tau_{x}^{k}\right)$$

#### 3.2.2. Gradient Ascent Methods

Alternatively one can try maximize the objective estimator $\hat{J}\left(\theta |p_{u^{⋆}}\right)$ using a gradient ascent method. Such a strategy is suited for general policy parametrizations that aim to find a global approximation of $u_{\mathrm{LSOC}}^{*}\left(t,x\right)$.
$$\theta_{g+1}=\theta_{g}+\eta \;\cdot \;\nabla_{\theta }{\hat{J}}_{g}\left(\theta |p_{u^{⋆}}\right)$$

Again regarding the choice of the goal distribution $p_{u^{⋆}}$ either the Path Integral Cross Entropy (PICE) or the Adaptive Smoothing Path Integral Control (ASPIC) method are obtained.

##### Path Integral Cross Entropy Method

PICE can be considered the gradient ascent version of SEPIC and was proposed in [41].
$$p_{u^{⋆}}=p_{u_{\mathrm{LSOC}}^{*}}$$

##### Adaptive Smoothing Path Integral Control (ASPIC) Method

ASPIC can be considered the gradient ascent version of PIREPS and was proposed more recently in [49].
$$\begin{array}{ccc} p_{u_{g+1}} & =\arg\min_{p_{u}\in P}E_{p_{u}}\left[L\right]+\lambda D\left[p_{u}\|p_{u_{g}}\right]p_{u^{⋆}} & =p_{u_{g+1}}\propto p_{u_{g}}^{\frac{\lambda }{1+\lambda }}\;\cdot \;p_{0}^{\frac{1}{1+\lambda }}\;\cdot \;e^{-\frac{1}{1+\lambda }\tilde{L}} \end{array}$$

### 3.3. Other Noteworthy PIC Methods

An important subset of PIC methods, or methods that are associated to the framework of LSOC at least, are Path Integral Policy Improvement algorithms or PI^2^. These methods are hard to classify as they are somewhere in between Evolutionary Search methods and policy optimization methods and rely on a heuristic temporal averaging strategy to resolve the conflicting policy parameter update schemes. The most important members of this class are PI^2^ [45], PI^2^ with Covariance Matrix Adaptation [25] and PI^2^ with Population Adaptation [50]. The authors of PI^2^-CMA, were the first to pursue the structural equivalence between Evolutionary Strategies and PIC methods. Based on this equivalence they proposed to adapt the policy covariance in correspondence to the CMA-ES algorithm, improving the convergence but destroying the underlying assumption of LSOC. This idea was successfully repeated in the context of trajectory optimization by [29].

### 3.4. Other Remarks

An issue of widespread concern in RL is how to shape a deliberately stochastic policy to obtain an explorative incentive without compromising safety measures or becoming risk seeking by unfortunate coincidence. PIC-based methods were one of the first strategies that somewhat address this issue with the implied connection between the uncertainty and cost. Especially, in case of deterministic systems where a deliberate stochastic policy could be introduced. Unfortunately the underlying theory of LSOC enforces an inverse proportionality between the control noise magnitude and the control penalty and moreover requires the control to be penalized quadratically (see later). Although justifiable from a control engineering perspective, this also poses severe practical limitations. Nowadays, entropy regularization seems to be a fruitful resolution to address the issue of stochastic policy shaping in a less restrictive or predetermined fashion. As for now it is not really clear how PIC methods relate to the framework of entropy regularized RL and whether they can benefit from recent advances made by this community. We shall address this question in the following section.

## 4. Entropy Regularized Path Integral Control

The main purpose of this section is to identify a novel SOC problem that can be solved explicitly yielding a formal optimal state trajectory distribution. The problem formulation is in that sense similar to the LSOC setting, yet it addresses a less restricted set of optimal control problems. Based on the general design principle for PIC methods discussed in Section 3.2, this optimal distribution can be substituted for the goal distribution $p_{u^{⋆}}$ hinting at a generalized set of PIC methods which will be investigate structurally for the purpose of model-free trajectory optimization in Section 5.

Our problem formulation relies heavily on the concept of entropy regularization. Entropy regularization is a setting of widespread use in the context of RL nowadays, yet it is less studied in the context of (static) stochastic optimization or stochastic search, at least in the way that we will treat it.

As we will show our SOC framework shares properties with static stochastic optimization and therefore the results that we derive in the latter setting will also hold in the context that enjoys our interest. Furthermore, we note that our results related to static stochastic optimization may be of interest to the Evolutionary Search community. Vice versa, the methods in Section 5 may benefit from prevailing ideas in the stochastic optimization community in order to deal with the problem of sample efficiency which is still considered to be a major challenge by the RL community. An example of such a strategy that is potentially interesting is importance mixing [51,52].

### 4.1. Entropy Regularized Optimization

There exists a large body of work that addresses the relation between inference and control. A lesser amount of work investigates the relation between inference and optimization. In this brief section, we provide an original and convincing argument to introduce an entropic regularization term into formal stochastic optimization problems that does not rely on an information-theoretic but on a strictly inference related argument. Although the resulting framework and associated distributions are known, our justification is original and in our opinion more intuitive. The concept results into a distribution sequence which exhibits a number of interesting properties and allows to make formal statements about convergence rates of derived practical search methods. As it will turn out these properties also directly apply to the entropy regularized optimal control problem that we will introduce in Section 4.3. To appreciate the argument, we must give an introduction to entropic inference.

#### 4.1.1. Entropic Inference

Inductive inference refers to the problem of how a rational agent should update its state of knowledge, or so-called belief, about some quantity when new information about that quantity becomes available. Beliefs about the quantity $x\in X\subset R^{n}$ are modeled as distributions. An inference procedure refers to the computational framework that establishes how to integrate new information with information held by any prior belief, say ρ, to determine an informed posterior, say π, consistently.

A well-known framework is that of Bayesian inference which allows to process new information in the form of data. A lesser known inference framework deals with the setting where information is available in the form of a constraint on the family of acceptable posteriors [53]. Specifically constraints in the form of the expected value of some function $f:X\to R^{n}$, i.e., $E_{}\left[f\right]=\mu$. Consequently we can focus our attention to the subset of distributions that agree with it $C_{f}\left(\mu \right):=\left\{\pi \in P:E_{\pi }\left[f\right]=\mu \right\}$. Any $\pi \in C_{f}\left(\mu \right)$ that satisfies the information constraint qualifies as a potential posterior belief. The challenge thus reduces to identifying a unique posterior from among all those that could give rise to the constraint. The solution is to establish a ranking on the set $C_{f}\left(\mu \right)$ by determining a functional F that associates a value to any posterior π relative to a given prior ρ. The inference procedure is then effectively cast into an optimization problem.
$$\pi^{*}=\arg\min_{\pi \in C_{f}\left(\mu \right)}F\left[\pi ,\rho \right]$$

The problem remains in finding a suitable and meaningful functional F. The measure should promote a distribution that is maximally unbiased (i.e., maximally ignorant) about all the information we do not possess. This principle is referred to as *minimal updating*. This problem setting roots back at least to the maximum entropy principle first introduced by Jaynes [54,55]. As many authors provided compelling theoretical arguments for the relative entropy as the only consistent measure [39,56,57,58,59,60], here it is important to emphasize that no interpretation of the measure is given. It simply is the only measure that agrees with the axioms of minimal updating.

The following variational problem determines the framework of entropic inference.
(11)$$\pi^{*}=\arg\min_{\pi \in P}D\left[\pi \|\rho \right]\;s.t.\;E_{\pi }\left[f\right]=\mu$$

We now possess of a rational argument as to why an entropy regularization term is added to any problem and what its effect will be on the solution.

#### 4.1.2. Optimization as an Inference Problem

Let us now argue how the principle of entropic inference can be practiced to serve the purpose of static or classic optimization. For convenience we assume that the objective $f:X\subset R^{n}\mapsto R$ has a unique global minimum.
$$x^{*}=\arg\min_{x\in X}f\left(x\right)$$

Assume we can model any beliefs we have about the solution $x^{*}$ by some prior distribution, say ρ∈P. Second, instead of supposing information in the form of the expected value of the objective *f*, here we only require that the expected value with respect to the posterior, π, is, some amount Δ>0, smaller than the expected value is with respect to the prior. In this fashion, we change our prior belief about the optimal solution but only to the minimal extent required to decrease the expectation taken over *f* with some arbitrary value Δ. Put differently, we obtain a posterior that makes least claim to being informed about the optimal solution beyond the stated lower limit on the expectation. This idea can be formalized accordingly
(12)$$\min_{\pi \in P}D\left[\pi \|\rho \right]\;s.t.\;E_{\pi }\left[f\right]+\Delta \leq E_{\rho }\left[f\right]$$

As the relative entropy minimizes for π=ρ, it follows that the inequality tightens into an equality. Nonetheless, we should be careful when we pick a value for Δ>0 that respects the bound $\Delta \leq E_{\rho }\left[f\right]-f^{*}$. This is a practical concern that does not interfere with what we wish to accomplish, which is to construct a minimal update procedure that we can practice to serve the purpose of optimization. It suffices to solve the problem above for π using variational calculus. This yields the following entropic update rule where λ>0 denotes the Lagrangian multiplier associated with the inequality constraint. This update principle is well known and can be subjected to an interesting interpretation (The posterior distribution, π, is equal to the prior distribution, ρ, multiplied with a cost driven probability shift, $e^{-\lambda f}$, that makes rewarding regions more probable, resembling the concept of Bayesian inference. The transformation $p\left(f\right)\propto e^{-\lambda f}$ maps costs to probabilities. Indeed one may recognize the inverse log-likelihood transformation from probability to cost as it is often used in the context of Bayesian inference) However, as far we are aware of, it has never been derived from the theory of entropic inference or thus minimal updating, which makes it possible to appreciate it in a more general light.
$$\pi \propto \rho \;\cdot \;e^{-\lambda f}$$

The exact value of λ can be determined by solving the dual optimization problem. Alternatively, we could also pick any λ>0 without the risk of overshooting the constraint $\Delta \leq E_{\rho }\left[f\right]-f^{*}$. We will show that when λ→∞, the expectation $E_{\pi }\left[f\right]$ collapses on the exact solution $f^{*}$. Therefore, $\frac{1}{\lambda }$ reduces to a *temperature* like quantity that determines the amount of information that is added to ρ, where in the limit exactly so much information is admitted to precisely determine $f^{*}$ and thus $x^{*}$.

The same strategy is obtained by considering the entropy regularized stochastic optimization problem below. Here an information-theoretic trust-region is introduced to promote conservative distribution updates but apart from the fact that the problem can be solved exactly, there is no proper motivation, at least not in the sense of minimal updating. This strategy has been adopted by previous authors from the optimization community [11,61].
$$\min_{\pi \in P}E_{\pi }\left[f\right]\;s.t.\;D\left[\pi \|\rho \right]\leq \Delta$$

Again it is easier to choose a suitable value for the implied Lagrangian multiplier λ>0 than it is for Δ>0. The problem above is therefore often relaxed using a penalty function.
(13)$$\pi^{*}=\arg\min_{\pi \in P}E_{\pi }\left[f\right]+\lambda D\left[\pi \|\rho \right]\propto \rho \;\cdot \;e^{-\frac{1}{\lambda }f}$$

#### 4.1.3. Theoretical Search Distribution Sequences

The entropic updating procedure suggested in (Equation 13) can be solved exactly and implies a theoretical distribution sequence, substituting the previous posterior for the next prior.
(14)$$\begin{array}{cc} \pi_{g+1} & \propto \pi_{g}\;\cdot \;e^{-\frac{1}{\lambda }f}=\min_{\pi \in P}E_{\pi }\left[f\right]+\lambda D\left[\pi \|\pi_{g}\right] \end{array}$$
(15)$$\begin{array}{cc} \pi_{g} & \propto \pi_{0}\;\cdot \;e^{-\frac{g}{\lambda }f} \end{array}$$

The update mechanism in (Equation 14) can be used as a theoretical model to shape practical search distribution sequences that can be used to solve the underlying optimization problem. In practice one seeks algorithms that estimate the posterior distribution $\pi_{g+1}$ from samples taken with the prior $\pi_{g}$. The theory now implies that the sequence will get gradually more informed about the optimum.

As far as we are aware of, the properties of the theoretical distribution sequence in (Equation 15) have not been studied before. In the following Theorem we summarize some of its properties. For the proof we refer to the work in Appendix A.

**Theorem** **2.**
*Assume that, without loss of generality, objective f attains a unique global minimum at the origin. Further define the sequence of search distributions $\left\{\pi_{g}\right\}$ so that $\pi_{g}\propto \pi_{0}\;\cdot \;e^{-gf}$. Then, it holds that*

*the distribution sequence $\left\{\pi_{g}\right\}$ collapses in the limit on the Dirac delta distribution in the sense that*
(16)$$\lim_{g\to \infty }\pi_{g}=\left\{\begin{array}{cccc} \; & \infty , & \; & x=x^{*}\\ \; & 0, & \; & x\neq x^{*} \end{array}\right.$$

*the function $E_{\pi_{g}}\left[f\right]$ of g is monotonically decreasing regardless of $\pi_{0}$*
(17)$$E_{\pi_{g+1}}\left[f\right]<E_{\pi_{g}}\left[f\right]$$

*the function $H\left[\pi_{g}\right]$ of g is monotonically decreasing if $\pi_{0}$ is chosen uniform on X*
(18)$$H\left[\pi_{g+1}\right]<H\left[\pi_{g}\right]$$



The property (Equation 17) is suggested by the problem definition in (Equation 12). It follows that the the sequence $\left\{E_{\pi_{g}}\left[f\right]\right\}$ converges monotonically to $f^{*}$, implying that $\left\{\pi_{g}\right\}$ should converge to $x^{*}$ (Equation 16). If we thus construct a stochastic optimization algorithm that maintains a sequence of search distributions governed by (Equation 14), the algorithm should converge monotonically to the optimal solution.

This sequence of search distributions is increasingly more informed about the solution and as a result its entropy content deteriorates as the sequence converges to the minimum (see Appendix A).

Properties (Equation 16) and (Equation 18) thus imply that the entropy slowly evaporates. This is an important observation indicating that the explorative incentive of the search distribution deteriorates for increasing *g*. Therefore, when we would use this sequence to shape a practical search algorithm, this property suggest that the entropy of the sample population will slowly diminish. As a result of the finiteness of the population it will become increasingly more likely that the correct distribution parameters cannot be inferred from the population so that the sequence collapses prematurely.

In order to stimulate the sequence to preserve an explorative incentive, i.e., manage the entropic content, problem (Equation 13) can be relaxed using an entropy term (γ>0), which will force the distribution to maintain a finite entropy.
(19)$$\pi_{g+1}=\arg\min_{\pi \in P}E_{\pi }\left[f\right]+\lambda D\left[\pi \|\pi_{g}\right]-\gamma H\left[\pi \right]$$

The entropy regularized problem above implies the following distribution sequence. This is easily verified relying on the calculus of variations.
(20)$$\begin{array}{cc} \pi_{g+1} & \propto \pi_{g}^{\frac{\lambda }{\lambda +\gamma }}\;\cdot \;e^{-\frac{1}{\lambda +\gamma }f} \end{array}$$
(21)$$\begin{array}{cc} \pi_{g} & \propto \pi_{0}^{{\left(\frac{\lambda }{\lambda +\gamma }\right)}^{g}}\;\cdot \;e^{-\frac{1}{\gamma }\left(1-{\left(\frac{\lambda }{\lambda +\gamma }\right)}^{g}\right)f}\propto \pi_{0}^{{\left(\frac{\lambda }{\lambda +\gamma }\right)}^{g}}\;\cdot \;\pi_{\infty }^{1-{\left(\frac{\lambda }{\lambda +\gamma }\right)}^{g}} \end{array}$$

As opposed to (Equation 15), sequence (Equation 21) does not collapse on the minimum, but instead converges to $\pi_{\infty }\propto e^{-\frac{1}{\gamma }f}$. The latter is equal to the exact solution of the entropy relaxed optimization problem defined below.
$$\pi_{\infty }=\arg\min_{\pi \in P}E_{\pi }\left[f\right]-\gamma H\left[\pi \right]$$

The derivative of the implied functions $E_{\pi_{g}}\left[f\right]$ and $H\left[\pi_{g}\right]$ are given by
(22)$$\begin{array}{cc} \frac{d}{dg}E_{\pi_{g}}\left[f\right] & =-\gamma \alpha \log\left(\frac{\lambda +\gamma }{\lambda }\right){\mathrm{Cov}}_{\pi_{g}}\left[\log\pi_{\infty },\log\frac{\pi_{\infty }}{\pi_{0}}\right] \end{array}$$
(23)$$\begin{array}{cc} \frac{d}{dg}H\left[\pi_{g}\right] & =-\alpha \log\left(\frac{\lambda +\gamma }{\lambda }\right)\left({\mathrm{Cov}}_{\pi_{g}}\left[\log\pi_{\infty },\log\frac{\pi_{\infty }}{\pi_{0}}\right]-\alpha {\mathrm{Var}}_{\pi_{g}}\left[\log\frac{\pi_{\infty }}{\pi_{0}}\right]\right) \end{array}$$
where $\alpha \left(g\right)={\left(\frac{\lambda }{\lambda +\gamma }\right)}^{g}>0$ so that $\lim_{g\to \infty }\alpha \left(g\right)=0$.

The convergence rate of the implied function $E_{\pi_{g}}\left[f\right]$ now also depends on the interaction between *f* and $\pi_{0}$. If the initial distribution $\pi_{0}$ is taken to be uniform on X, so that ${\mathrm{cov}}_{\pi_{g}}\left[\pi_{0},f\right]=0$, the expected value will decrease monotonically. As opposed to (Equation 18) the rate will stall for large *g*. Derivations are provided in Appendix B.

In Section 4.3, we introduce a stochastic optimal control problem that is characterized by a similar distribution sequence. Therefore it inherits the properties described above. The usefulness of these theoretical distributions, is illustrated in Appendix C where we derive a stochastic search method.

Next, let us return to our original problem and briefly review entropy regularized SOC and its usefulness to RL as a stepping stone to Section 4.3.

### 4.2. Entropy Regularized Optimal Control

As we already explained in the introduction, there is no reason to maintain a stochastic policy in the generic SOC setting (Equation 2). On the other hand, a stochastic policy can be of interest to derive policy updating algorithms that depend on stochasticity for improved exploration rather than to rely on the stochastic system dynamics only.

As was justified rigorously in the previous section this can be achieved by introducing an entropy relaxation term in the optimization objective. Such a term actively stimulates a stochastic policy. Nowadays, this is a setting of widespread use in the RL community and was explored extensively by Levine et al. particularly in the context of Actor-Critic RL algorithms [35,62]. In recent literature, this problem has been regularized successfully using a relative entropy relaxation term in order to limit policy oscillations between updates [36,37,38,63,64]. These references address the following entropy regularized SOC problem.
(24)$$\begin{array}{cc} V_{g+1}\left(t,x\right) & =\min_{\pi \in P}E_{\pi }\left[r_{t}\left(x,u\right)+E_{p}\left[V_{g+1}\left(t+1,x^{\prime }\right)\right]\right]-\gamma H\left[\pi \right]+\lambda D\left[\pi \|\pi_{g}\right] \end{array}$$
(25)$$\begin{array}{cc} \pi_{g+1}\left(u|t,x\right) & =\arg\min_{\pi \in P}E_{\pi }\left[r_{t}\left(x,u\right)+E_{p}\left[V_{g+1}\left(t+1,x^{\prime }\right)\right]\right]-\gamma H\left[\pi \right]+\lambda D\left[\pi \|\pi_{g}\right] \end{array}$$

Relying on the calculus of variations, one can verify that the solution is given by [38,65]
$$\begin{array}{cc} V_{g+1} & =\left(\lambda +\gamma \right)\log\int \pi_{g}^{\frac{\lambda }{\lambda +\gamma }}e^{-\frac{1}{\lambda +\gamma }\left(r+E_{p}\left[V_{g+1}^{\prime }\right]\right)}du\\ \pi_{g+1} & \propto \pi_{g}^{\frac{\lambda }{\lambda +\gamma }}e^{-\frac{1}{\lambda +\gamma }\left(r+E_{p}\left[V_{g+1}^{\prime }\right]\right)} \end{array}$$

The authors in the respective references do use this problem formulation as a starting point to derive policy search methods. Because of the expectation in the exponent, it is impossible to practice the recursion and find an explicit expression for the corresponding optimal path distribution which would circumvent the dependency on the value function. Therefore all of them require to estimate the value, $V_{g}\left(t,x\right)$, or state-action value function, $Q_{g}\left(t,x,u\right)$, either using a general function approximator or a local approximation. As a consequence, the entropy regularized optimal control framework can also not be used to derive a PIC method, at least not in the sense discussed in Section 3.2.

In this article, we specifically limit our focus to the class of PIC methods described in Section 3.2, which as stated assume the existence of an explicit goal state trajectory distribution. In the following section, we introduce an adjusted entropy regularized stochastic optimal problem for which there does exists a formal yet explicit optimal path distribution, and that consequently can be treated with the PIC machinery put in place.

### 4.3. Entropy Regularized Trajectory Optimization

As discussed in the previous section, there exists no explicit expression for the optimal path distribution sequence in the setting of entropy regularized SOC. Consequently, the framework can not be leveraged by the PIC design principle described in Section 3.2. Here, we aim to identify an entropy regularized SOC problem that can be solved for an explicit trajectory distribution. As was identified to be an essential condition in the setting of LSOC, therefore we must get rid of the stochastic policy, π. Therefore, we suggest to absorb it into the implied state transition distribution, $p_{\pi }$. Analogously, this will elevate the problem from the policy to the state transition distribution optimization space, provided that we also can get rid of the dependency of the cost rate $r_{t}\left(x,u\right)$ on the control effort *u*.

Let us therefore assume there exists an inverse dynamic function $f^{-1}$ so that $u=f^{-1}\left(t,x,x^{\prime }\right)$ when $x^{\prime }=f\left(t,x,u\right)$. This implies a trajectory cost rate function $c_{t}$ that accumulates into a trajectory cost $C=\sum_{t=0}^{T}c_{t}$. Specifically, we have that $c_{t}\left(x,x^{\prime }\right)=r_{t}\left(x,f^{-1}\left(x,x^{\prime }\right)\right)$ and $c_{T}\left(x\right)=r_{T}\left(x\right)$ (We emphasize here that the existence of such an inverse model is a mere formal assumption. In the practical setting to be presented in Section 5, we will never actually have to invert the system dynamics given that we have direct access to the control values that have been applied to the system).

These definitions allow to define a stochastic state trajectory optimization problem. We can rewrite the stochastic Bellman Equation (Equation 2) as follows,
$$V\left(t,x\right)=\min_{\pi }\int \int \left(r_{t}\left(x,u\right)+V\left(t+1,x^{\prime }\right)\right)p\left(x^{\prime }|x,u\right)\pi \left(u|t,x\right)dudx^{\prime }$$
and then substitute the expressions above. This yields a similar problem
$$V\left(t,x\right)=\min_{\pi }\int \left(c_{t}\left(x,x^{\prime }\right)+V\left(t+1,x^{\prime }\right)\right)p_{\pi }\left(x^{\prime }|t,x\right)dx^{\prime }$$

As there is a one-to-one correspondence between π and $p_{\pi }$, one can replace the minimization with respect to the policy by a minimization with respect to the implied state transition distribution $p_{\pi }$. Finally, we can regularize and relax the problem according to the general entropic principles justified in Section 4.1 and as are thus also common in the RL community.

We end up with what we refer to as the Entropy Regularized Trajectory Optimization (ERTO) problem.
(26)$$\begin{array}{cc} V_{g+1}\left(t,x\right) & =\min_{p_{\pi }\in P}E_{p_{\pi }}\left[c_{t}\left(x,x^{\prime }\right)+V_{g+1}\left(t+1,x^{\prime }\right)\right]-\gamma H\left[p_{\pi }\right]+\lambda D\left[p_{\pi }\|p_{\pi_{g}}\right] \end{array}$$
(27)$$\begin{array}{cc} p_{\pi_{g+1}}\left(x^{\prime }|t,x\right) & =\arg\min_{p_{\pi }\in P}E_{p_{\pi }}\left[c_{t}\left(x,x^{\prime }\right)+V_{g+1}\left(t+1,x^{\prime }\right)\right]-\gamma H\left[p_{\pi }\right]+\lambda D\left[p_{\pi }\|p_{\pi_{g}}\right] \end{array}$$

Remark that this problem is fundamentally different from that given in (Equation 24) considering that the penalties, $D\left[\pi \|\pi_{g}\right]$ and $D\left[p_{\pi }\|p_{\pi_{g}}\right]$, and, $H\left[\pi \right]$ and $H\left[p_{\pi }\right]$, do not express the same restrictions.

Analogously to the LSOC, we summarize the most profound properties of problem (Equation 26) in the following theorem. The theorem also establishes a relation between the optimal stochastic policy $\pi_{g}\left(t,x\right)$, the optimal state transition distribution $p_{\pi_{g}}\left(x^{\prime }|t,x\right)$ and the optimal state trajectory distribution $p_{\pi_{g}}\left(\tau_{x}\right)$. Again, the proof relies on the calculus of variations mostly. We direct the reader to Appendix D for details.

**Theorem** **3.**
*With $V_{g+1}\left(t,x\right)$ defined as in (Equation 26), the following problems are equivalent,*
(28)$$\begin{array}{cc} \pi_{g+1}\left(u|t,x\right) & =\arg\min_{\pi \in P}E_{p_{\pi }}\left[c_{t}\left(x,x^{\prime }\right)+V_{g+1}\left(t+1,x^{\prime }\right)\right]-\gamma H\left[p_{\pi }\right]+\lambda D\left[p_{\pi }\|p_{\pi_{g}}\right] \end{array}$$
(29)$$\begin{array}{cc} p_{\pi_{g+1}}\left(x^{\prime }|t,x\right) & =\arg\min_{p_{\pi }\in P}E_{p_{\pi }}\left[c_{t}\left(x,x^{\prime }\right)+V_{g+1}\left(t+1,x^{\prime }\right)\right]-\gamma H\left[p_{\pi }\right]+\lambda D\left[p_{\pi }\|p_{\pi_{g}}\right] \end{array}$$
(30)$$\begin{array}{cc} p_{\pi_{g+1}}\left(\tau_{x}\right) & =\arg\min_{p_{\pi }\in P}E_{p_{\pi }}\left[C\right]-\gamma H\left[p_{\pi }\right]+\lambda D\left[p_{\pi }\|p_{\pi_{g}}\right] \end{array}$$

*The latter can be solved explicitly,*
$$\begin{array}{cc} p_{\pi_{g+1}}\left(x^{\prime }|t,x\right) & =p_{\pi_{g}}{\left(x^{\prime }|t,x\right)}^{\frac{\lambda }{\lambda +\gamma }}e^{-\frac{1}{\lambda +\gamma }\left(c_{t}\left(x,x^{\prime }\right)+V\left(t+1,x^{\prime }\right)\right)}\\ p_{\pi_{g+1}}\left(\tau_{x}\right) & =p_{\pi_{g}}{\left(\tau_{x}\right)}^{\frac{\lambda }{\lambda +\gamma }}e^{-\frac{1}{\lambda +\gamma }C\left(\tau_{x}\right)} \end{array}$$

*where $V_{g+1}\left(t,x\right)$ is governed by the recursion*
$$V_{g+1}\left(t,x\right)=\left(\lambda +\gamma \right)\log\int p_{\pi_{g}}{\left(x^{\prime }|t,x\right)}^{\frac{\lambda }{\lambda +\gamma }}e^{-\frac{1}{\lambda +\gamma }\left(c_{t}\left(x,x^{\prime }\right)+V\left(t+1,x^{\prime }\right)\right)}dx^{\prime }$$

*and where $\pi_{g}\left(u|t,x\right)$ and $p_{\pi_{g}}\left(x^{\prime }|t,x\right)$ are related as*
$$p_{\pi_{g}}\left(x^{\prime }|t,x\right)=\int p\left(x^{\prime }|x,u\right)\pi_{g}\left(u|t,x\right)du$$


We list the most important implications of Theorem 3:As a result of the entropy regularization and the state trajectory lifted optimization space, it is now possible to obtain another formal yet explicit optimal state trajectory distribution. Note that this is not the same optimal distribution as we derived in the LSOC setting given that here the control is not penalized through a Kullback–Leibler divergence term but is penalized implicitly through the cost *C*. When we evaluate *C* and have access to the state-action trajectory τ, we can simply replace *C* by *R*. Here, we emphasize that $p_{\pi_{g}}$ still represents the state trajectory distribution which is now also a function of the actions.
(31)$$p_{\pi_{g+1}}\propto p_{\pi_{g}}^{\frac{\lambda }{\lambda +\gamma }}\;\cdot \;e^{-\frac{1}{\lambda +\gamma }C}\equiv p_{\pi_{g}}^{\frac{\lambda }{\lambda +\gamma }}\;\cdot \;e^{-\frac{1}{\lambda +\gamma }R}$$Second, the theorem implies that we can readily apply the PIC design strategy described in Section 3.2, substituting the optimal path distribution sequence (Equation 31) for $p_{\pi }^{⋆}$, and, the parametrized path distribution $p_{\pi (\theta )}$ induced by some stochastic parametric policy π(θ) for $p_{u(\theta )}$, to derive a generalized class of PIC methods. We refer to this class as Entropy Regularized Path Integral Control or ERPIC.
(32)$$\begin{array}{cc} \min_{\theta }D\left[p_{\pi_{g+1}}\|p_{\pi (\theta )}\right] & \propto E_{p_{\pi (\theta_{g})}}\left[\frac{p_{\pi_{g+1}}}{p_{\pi (\theta_{g})}}\log p_{\pi (\theta )}\right]+\mathrm{constant}\\ \; & \approx \frac{1}{M}\sum_{k=1}^{M}{p_{\pi (\theta_{g})}\left(\tau_{x}^{k}\right)}^{-\frac{\gamma }{\lambda +\gamma }}\;\cdot \;e^{-\frac{1}{\lambda +\gamma }C\left(\tau_{x}^{k}\right)}\;\cdot \;\log p_{\pi (\theta )}\left(\tau_{x}^{k}\right)+\mathrm{constant} \end{array}$$As their is no longer a formal difference between problem (Equation 19) and (Equation 30), it follows that the properties derived for the sequence (Equation 21) also hold for the optimal state trajectory distribution sequence $\left\{p_{\pi_{g}}\right\}$ and imply monotonic convergence to $p_{\pi_{\infty }}\left(\tau_{x}\right)\propto \exp\left(-\frac{1}{\gamma }C\left(\tau_{x}\right)\right)$.

We conclude this section with a final remark related to control affine deterministic system dynamics, $x^{\prime }=a\left(x\right)+B\left(x\right)u$, in this setting controlled by a stochastic policy of the form $\pi_{g}\left(u|t,x\right)=N\left(u|u_{g}\left(t,x\right),\Sigma_{g}\left(t,x\right)\right)$. Again, it is possible to derive an explicit expression for the optimal policy. The controlled transition distribution is given by $p\left(x^{\prime }|t,x,u\right)=N\left(x^{\prime }|a\left(x\right)+B\left(x\right)u_{g}\left(t,x\right),B\left(x\right)\Sigma_{g}\left(t,x\right)B{\left(x\right)}^{⊤}\right)$. By matching the moments of the two distributions $N(x^{\prime }|a\left(x\right)+B\left(x\right)u_{g+1}\left(t,x\right),B\left(x\right)\Sigma_{g+1}\left(t,x\right)B{\left(x\right)}^{⊤})$ and $p_{\pi_{g+1}}\left(x^{\prime }|t,x\right)$ we find expressions for $u_{g+1}\left(t,x\right)$ and $\Sigma_{g+1}\left(t,x\right)$.

It is easily verified that
(33)$$u_{g+1}\left(t,x\right)=u_{g}\left(t,x\right)+\frac{E_{p_{\pi_{g}}\left(\tau_{x}|t,x\right)}\left[\xi \;\cdot \;e^{-\frac{1}{\lambda +\gamma }C\left(\tau_{x}|t,x\right)}\right]}{E_{p_{\pi_{g}}\left(\tau_{x}|t,x\right)}\left[e^{-\frac{1}{\lambda +\gamma }C\left(\tau_{x}|t,x\right)}\right]}$$
and
(34)$$\begin{array}{c} \Sigma_{g+1}\left(t,x\right)=\frac{E_{p_{\pi_{g}}\left(\tau_{x}|t,x\right)}\left[\xi \xi^{⊤}\;\cdot \;e^{-\frac{1}{\lambda +\gamma }C\left(\tau_{x}|t,x\right)}\right]}{E_{p_{\pi_{g}}\left(\tau_{x}|t,x\right)}\left[e^{-\frac{1}{\lambda +\gamma }C\left(\tau_{x}|t,x\right)}\right]}\\ -\frac{E_{p_{\pi_{g}}\left(\tau_{x}|t,x\right)}\left[\xi \;\cdot \;e^{-\frac{1}{\lambda +\gamma }C\left(\tau_{x}|t,x\right)}\right]}{E_{p_{\pi_{g}}\left(\tau_{x}|t,x\right)}\left[e^{-\frac{1}{\lambda +\gamma }C\left(\tau_{x}|t,x\right)}\right]}\frac{E_{p_{\pi_{g}}\left(\tau_{x}|t,x\right)}\left[\xi^{⊤}\;\cdot \;e^{-\frac{1}{\lambda +\gamma }C\left(\tau_{x}|t,x\right)}\right]}{E_{p_{\pi_{g}}\left(\tau_{x}|t,x\right)}\left[e^{-\frac{1}{\lambda +\gamma }C\left(\tau_{x}|t,x\right)}\right]} \end{array}$$
where $\xi ∼N(0,\Sigma_{g}\left(t,x\right))$.

We point out the similarities with (Equation 7). Further note that for λ=0, the information-theoretic trust-region is lifted in which case (Equation 33) can be applied in the same MPC sense as (Equation 7).

## 5. Formal Comparison of Path Integral Control Methods

In this final section, we give a formal comparison of three different PIC methods based on the general design principle identified in Section 3.2 that are tailored to trajectory optimization. The goal distribution $p_{u^{⋆}}$ can now either root from the LSOC or the ETRO setting, respectively, introduced in Section 3.1 and Section 4.3. The methods we discuss are tailored to find local solutions. That is policies that will control the system when the initial state is equal to a given state $x_{0}$. Sometimes on refers to such strategies as trajectory-based policy optimization methods. All of the methods are model-free and sample-based. Currently, such methods are used in the literature either to deal with complex simulation environments where traditional gradient based trajectory optimizers fall short [28,42] or to derive reinforcement learning algorithms [29,41,49].

The methods discussed here are in that sense closest related to [38]. However, this algorithm derives from the generic entropy regularized SOC problem (Equation 24) which differs significantly from the problem formulation proposed in Section 4.3. We come back to this later.

Remark that depending on the theoretical setting, either LSOC or ETRO, we have to use a different parametrized state trajectory distribution which has an effect on the PIC objectives. In case of LSOC based PIC we address the following objective,
(35a)$${\hat{J}}_{g,\mathrm{LSOC}}\left(\theta \right)=\sum_{k=1}^{K}\frac{w_{g}^{k}}{\sum_{k=1}^{K}w_{g}^{k}}\log p_{u(\theta )}\left(\tau_{x}^{k}\right)$$
while in case of ERTO based PIC we address
(35b)$${\hat{J}}_{g,\mathrm{ERTO}}\left(\theta \right)=\sum_{k=1}^{K}\frac{w_{g}^{k}}{\sum_{k=1}^{K}w_{g}^{k}}\log p_{\pi (\theta )}\left(\tau_{x}^{k}\right)$$

The values $w_{g}^{k}$ differ for each methods and will be discussed in the following section.

### 5.1. Control Affine Systems

We concentrate on systems governed by deterministic control affine dynamics. In either case this is the sole condition for which an explicit solution exists for the optimal policies $u_{\mathrm{LSOC}}\left(t,x\right)$ or $\pi_{g}\left(t,x\right)$, given in (Equation 7) or (Equation 33)), respectively. Furthermore, it is also the sole condition for which $w_{g}^{k}$ can be evaluated exactly. This assumption introduces little practical limitations as many systems comply to this system model.
$$x^{\prime }=a\left(x\right)+B\left(x\right)u$$

Further let us assume Gaussian policies of the form
$$\pi \left(u|t,x\right)=N\left(u|u_{t}\left(x\right),\Sigma_{t}\left(x\right)\right)$$

In the case that we discuss methods derived from the LSOC setting, this means that we deliberately introduce a stochastic policy to mimic stochastic system dynamics. We emphasize that in this case $\Sigma_{t}\left(x\right)$ can not be chosen freely as it is directly related to the control effort penalization. From here on forth, we can thus consider stochastic policies in the LSOC setting as long as we silently acknowledge that the covariance can not be chosen freely.

Then the state transition distribution is given by
$$p_{\pi }\left(x^{\prime }|t,x\right)=N\left(x^{\prime }|a\left(x\right)+B\left(w\right)u_{t}\left(x\right),B\left(x\right)\Sigma_{t}\left(x\right)B{\left(x\right)}^{⊤}\right)$$

We can now substitute these expression into the different PIC objectives defined throughout the rest of the paper to find expressions for $w_{g}^{k}$.

Therefore, we will require the following intermediate results,
$$\begin{array}{cc} \log p_{0}\left(\tau_{x}^{k}\right) & =\sum_{t=0}^{T-1}\log p_{0}\left(x_{t+1}^{k}\left|t,x_{t}^{k}\right.\right)\propto -\sum_{t=0}^{T-1}\frac{1}{2}{∥u_{g,t}+\delta u_{t}^{k}∥}_{R_{t}^{k}}^{2}\\ \log p_{\pi (\theta_{g})}\left(\tau_{x}^{k}\right) & =\sum_{t=0}^{T-1}\log p_{\pi (\theta_{g})}\left(x_{t+1}^{k}\left|t,x_{t}^{k}\right.\right)\propto -\sum_{t=0}^{T-1}\frac{1}{2}{∥\delta u_{t}^{k}∥}_{R_{t}^{k}}^{2} \end{array}$$
where $R_{t}=B^{⊤}{\left(B\Sigma_{t}B^{⊤}\right)}^{-1}B\approx \Sigma_{t}^{-1}$ and $\delta u_{t}^{k}∼N\left(0,\Sigma_{t}\left(x_{t}^{k}\right)\right)$. These results follow from the fact that the samples satisfy the following condition
$$x_{t+1}^{k}=a_{t}^{k}+B_{t}^{k}u_{g,t}^{k}+B_{t}^{k}\delta u_{t}^{k}$$

Finally we can write the weights $w_{g}^{k}$ as $e^{-P(\tau_{x}^{k})}$ where the function P(·) depends on the specific objective. We will address two LSOC based objectives associated with SEPIC (or PICE) (Equation 9) and PIREPS (or ASPIC) (Equation 10) and the ETRO based objective associated to ERPIC (Equation 32). When we substitute the appropriate expressions in the corresponding objectives, we obtain, respectively,

SEPIC (or PICE)
$$P_{\mathrm{SEPIC}}=l_{T}\left(x_{T}\right)+\sum_{t=0}^{T-1}l_{t}\left(x_{t}\right)+\sum_{t=0}^{T-1}\frac{1}{2}{∥u_{g,t}+\delta u_{t}∥}_{R_{t}}^{2}-\sum_{t=0}^{T-1}\frac{1}{2}{∥\delta u_{t}∥}_{R_{t}}^{2}$$The function *P* reads as the cost accumulated over the trajectory $\tau^{k}$ where the states and control efforts are penalized separately. The trailing term is included to compensate for the full noise penalization [41]. To penalize the states any nonlinear function can be used, the control is penalized using a quadratic penalty term which depends on the noise added to the system. A crucial limitations is clearly that the exploration noise and the control cost are therefore coupled.PIREPS (or ASPIC)
$$P_{\mathrm{PIREPS}}=\frac{1}{1+\lambda }P_{\mathrm{SEPIC}}$$It turns out that in this setting the weights $w_{g}^{k}$ are simply a smoothed version of those associated to SEPIC. For λ≫1 (i.e., strong regularization) the weights will all have approximately the same value and therefore $u_{g+1}\left(t,x\right)\approx u_{g}\left(t,x\right)$. For 1≫λ>0 (i.e., weak regularization), the method reduces to SEPIC.ERPIC
$$P_{\mathrm{ERPIC}}=\frac{1}{\lambda +\gamma }r_{T}\left(x_{T}\right)+\frac{1}{\lambda +\gamma }\sum_{t=0}^{T-1}r_{t}\left(x_{t},u_{t}\right)-\frac{\gamma }{\lambda +\gamma }\sum_{t=0}^{T-1}\frac{1}{2}{∥\delta u_{t}∥}_{R_{t}}^{2}$$One can easily verify that in the ERPIC setting, function *P* represents the cost accumulated over the trajectory $\tau^{k}$ where the states and control efforts are no longer penalized separately. Here, a discount terms is included that promotes uncertain trajectories making sure that the entropy of the search distribution does not evaporate eventually. Second, we wish to point out the obvious similarities with the stochastic search method in Appendix C.

### 5.2. Locally Linear Gaussian Policies

In conclusion, we solve the optimization problem defined in (Equation 32) exactly. Therefore, we will approximate the Gaussian policy $N(u|u_{t}\left(x\right),\Sigma_{t}\left(x\right))$ with a locally linear Gaussian feedback policy of the form $N(u|u_{g,t}+K_{g,t}x,\Sigma_{g,t})$.

In the ERPIC setting, the proportionality between the control penalization and the injected noise is lifted and therefore we gain access to the full parametrization of the policies, specifically $\theta_{g,t}=\left\{a_{g,t},K_{g,t},\Sigma_{g,t}\right\}$. Starting from ([Disp-formula FD35b-entropy-22-01120]) it follows that
(36)$$\begin{array}{cc} \left\{a_{g,t},K_{g,t},\Sigma_{g,t}\right\} & =\max_{\theta_{g,t}}\sum_{k=1}^{K}\frac{w_{g}^{k}}{\sum_{k=1}^{K}w_{g}^{k}}\log p_{\pi (\theta )}\left(\tau_{x}^{k}\right)\\ \; & =\max_{\theta_{g,t}}\sum_{k=1}^{K}\frac{w_{g}^{k}}{\sum_{k=1}^{K}w_{g}^{k}}\sum_{t=0}^{T-1}\log N\left(u|u_{g,t}+K_{g,t}x,\Sigma_{g,t}\right)\\ \; & =\max_{\theta_{g,t}}\sum_{k=1}^{K}\frac{w_{g}^{k}}{\sum_{k=1}^{K}w_{g}^{k}}\log N\left(u|u_{g,t}+K_{g,t}x,\Sigma_{g,t}\right) \end{array}$$

This procedure then yields the following elaborate updates (we refer to Appendix E for a proper derivation). Notation 〈〈f〉〉 is shorthand for the likelihood weighted average $\sum_{k=1}^{K}\frac{w_{g}^{k}}{\sum_{k=1}^{K}w_{g}^{k}}f^{k}$ while 〈f〉 is shorthand for the empirical mean $\frac{1}{K}\sum_{k=1}^{K}f^{k}$.
(37)$$\begin{array}{cc} u_{g+1,t} & =u_{g,t}+\Delta {\hat{\mu }}_{u,g,t}-{\hat{\Sigma }}_{ux,g,t}{\hat{\Sigma }}_{xx,g,t}^{-1}\left({\hat{x}}_{g,t}+\Delta {\hat{\mu }}_{x,g,t}\right) \end{array}$$
(38)$$\begin{array}{cc} K_{g+1,t} & ={\hat{\Sigma }}_{ux,g,t}{\hat{\Sigma }}_{xx,g,t}^{-1} \end{array}$$
(39)$$\begin{array}{cc} \Sigma_{g+1,t} & ={\hat{\Sigma }}_{uu,g,t}-{\hat{\Sigma }}_{ux,g,t}{\hat{\Sigma }}_{xx,g,t}^{-1}{\hat{\Sigma }}_{xu,g,t} \end{array}$$
with
$$\begin{array}{cc} \Delta {\hat{\mu }}_{u,g,t} & =\langle \langle \;\Delta u_{t}^{k}\rangle \rangle \;\\ \Delta {\hat{\mu }}_{x,g,t} & =\langle \langle \;\Delta x_{t}^{k}\rangle \rangle \;\\ {\hat{\Sigma }}_{uu,g,t} & =\langle \langle \;\Delta u_{t}^{k}\Delta u_{t}^{k,⊤}\rangle \rangle \;\\ {\hat{\Sigma }}_{ux,g,t} & =\langle \langle \;\Delta u_{t}^{k}\Delta x_{t}^{k,⊤}\rangle \rangle \;\\ {\hat{\Sigma }}_{xx,g,t} & =\langle \langle \;\Delta x_{t}^{k}\Delta x_{t}^{k,⊤}\rangle \rangle \; \end{array}$$
where $\Delta x_{t}^{k}=x_{t}^{k}-{\hat{x}}_{g,t}$ with ${\hat{x}}_{g,t}=\langle x_{t}^{k}\rangle$ and $\Delta u_{t}^{k}=u_{t}^{k}-{\hat{u}}_{g,t}$ with ${\hat{u}}_{g,t}=\langle u_{t}^{k}\rangle =u_{g,t}+K_{g,t}{\hat{x}}_{g,t}$ so that $\Delta u_{t}^{k}=\delta u_{t}^{k}+K_{g,t}\Delta x_{t}^{k}$.

### 5.3. Discussion

As was made clear in the introduction of the paper, it is not our intention to provide a numerical analysis or study. We are merely concerned with the relation between all the topics that we touched upon. In conclusion, we will discuss therefore a number of observations that are of interest to fully grasp the relation between Path Integral Control methods, Stochastic Search methods and Reinforcement Learning and the implied limitations.

#### 5.3.1. Remarks Related to Stochastic Search Methods and Variance

In this section, we comment on Stochastic Search Methods from two perspectives: The first perspective is related to the entropy regularized optimization framework and the theoretical search distribution sequences introduced in Section 4.1. The second perspective is related to the apparent connections between Entropy Regularized Optimization tailored to standard optimization and Entropy Regularized Trajectory Optimization which is tailored to optimal control problems.

As far as we are aware of, we provide an original argument to address the standard optimization problem as a probabilistic inference problem. The idea of treating the belief about the solution as a distribution function and aiming to reduce the expected cost value with each belief update is to our opinion very intuitive. It seem to be ideas worth pursuing whether existing stochastic search methods fit this abstract framework and whether other practical methods can be modeled after the theoretical distribution sequences. However, the mathematical framework can also be rewritten as a variational optimization problem with respect to a search distribution regularized by information-theoretic or information-geometric trust-regions. From this perspective our entropic inference argument sheds a new light on earlier work in the context of stochastic search methods [11,12,14,36,61,64].

By conducting an Entropy Regularization in the context of Stochastic Optimal Control that draws direct inspiration from the LSOC setting, we were able to formulate ETRO. This problem shares the crucial trait with the LSOC problem that made it a topic of interest in learning-based control. As a result the problems looses the so called optimal substructure property which is characteristic to Optimal Control and can therefore be treated as a standard entropy regularized optimization problem. This condition has two consequences: The theoretical analysis become easier to handle. On the other hand, the framework infers the temporal policy parameters as if they influence the entire trajectory instead of only from time instant *t*.

Following the latter observation, it is interesting to compare the updates in (Equation 37) with those in Appendix C. In retrospect, these are almost identical apart from one crucial difference. The same weights $w_{g}^{k}$ are used to update the temporal parameters $\left\{u_{g,t},K_{g,t},\Sigma_{g,t}\right\}$ regardless of *t*. This relates directly to the comment above and has a dreadful practical implication. The effect of a random policy variations $\delta u_{t}^{k}$ at time *t* is accounted for by the complete associated trajectory. The trajectory itself is however influenced by random policy variations at different time instants $t^{\prime }\in \left\{0,\cdots ,t\;-\;1,t\;+\;1,\cdots ,T-1\right\}$. We remark that the approach is therefore expected to suffer from high variance, especially for T≫1, which will ultimately destabilize the convergence. This observation is a fundamental flaw of the framework described in Section 3.2 and therefore holds for any PIC methods including any existing methods. The high variance can partially be alleviated by acknowledging that the parameter update at time instant *t* is independent from the stochastic variables ${\left\{x_{t^{\prime }}^{k},u_{t^{\prime }}^{k}\right\}}_{t^{\prime }=0}^{t-1}$ and therefore the optimization problem in (Equation 36) can be revised using temporal weights
$$\max_{\theta_{g,t}}\sum_{k=1}^{K}\frac{w_{g,t}^{k}}{\sum_{k=1}^{K}w_{g,t}^{k}}\log N\left(u|u_{g,t}+K_{g,t}x,\Sigma_{g,t}\right)$$
where $w_{g,t}^{k}=w_{g,t}\left(\tau^{k}\right)=\exp\left(-P_{t,\mathrm{ERPIC}}\left(\tau^{k}\right)\right)$ with
$$P_{t,\mathrm{ERPIC}}\left(\tau \right)=\frac{1}{\lambda +\gamma }r_{T}\left(x_{T}\right)+\frac{1}{\lambda +\gamma }\sum_{t^{\prime }=t}^{T-1}r_{t}\left(x_{t^{\prime }},u_{t^{\prime }}\right)-\frac{\gamma }{\lambda +\gamma }\sum_{t^{\prime }=t}^{T}\frac{1}{2}{∥\delta a_{t^{\prime }}∥}_{R_{t^{\prime }}}^{2}$$

The update is then more in line with the explicit expression for the optimal stochastic policy that is given in (Equation 33).

#### 5.3.2. Positioning of PIC Methods within the Field of RL

Formerly it was unclear how the framework of Linearly Solvable Optimal Control was related exactly to other RL methods. By identifying the framework of Entropy Regularized Trajectory Optimization and comparing it with the framework of Entropy Regularized Stochastic Optimal Control this ambiguity is lifted. Second, the model-free (local) trajectory optimization algorithms presented in this section are closest related to the method presented in [38]. However, this algorithm is based on problem formulation (Equation 24) and therefore requires to estimate the $Q_{g}$-function which is not required in this strictly PIC-based setting. Related to the final comment above, this circumvents the problem of temporal variance induced by random policy variations at different time instants and is therefore expected to perform superiorly in practice.

## 6. Conclusions

In this paper, we addressed PIC methods, a class of policy search methods which have been closely tied to the theoretical setting of LSOC. Our work was motivated primarily by an unsatisfactory understanding and generality of Path Integral Control methods in relation with the setting of LSOC. Nevertheless, referred class of policy search methods enjoyed interest in the RL community with applications ranging from robust trajectory optimization and guided policy search to robust model based predictive control.

The LSOC setting was considered to be unique in the sense that an explicit expression exists for the optimal state trajectory distribution. The former we identified to be fundamental to the class of PIC methods. We illustrate that the existence of such a solution is not a unique trait to the setting of LSOC and argue that a similar solution can be derived from within the setting of Entropy Regularized Stochastic Optimal Control, a framework of widespread use nowadays in the RL community. The setting of Entropy Regularized Stochastic Optimal Control allows to treat a more general class of optimal control problems than the setting of LSOC, rendering LSOC obsolete.

In either case, the result follows from lifting the stochastic optimal control problem from the policy to the state transition distribution space making the control implicit. As a result of this characteristic, the properties of the implied state trajectory distribution sequence can be analyzed. We show that the sequence converges monotonically to the solution of the underlying deterministic optimal control problem.

To treat this sequence more formally, we give an original and compelling argument for the use of information-geometric measures in the context of stochastic search algorithms based on the principle of entropic inference. The main idea is to maintain a belief function over the solution space that is least committed to any assumption about the solution, apart from the requirement that the expectation over the objective should decrease monotonically between updates. The resulting Entropy Regularized Optimization framework may serve as an overarching paradigm to analyze and derive stochastic search algorithms.

In conclusion, the value of this article is in that it identifies PIC methods for what they really are, a class of model-free and approximation free policy search algorithms that are theoretically founded on a specific subclass of Entropy Regularized Stochastic Optimal Control. Therewith, the underlying machinery associated to preceding derivations is untied from the peculiar setting of LSOC and the relation between PIC methods and state-of-the-art Reinforcement Learning is finally demystified. Nevertheless, our investigation suggests that PIC methods in fact are structurally closer related to Stochastic Search Methods tailored to classic optimization problem than they are to state-of-the-art Reinforcement Learning methods tailored to optimal control and decision-making problems and that the associated policy parameter updates will therefore be prone to higher variances.

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
