# Peer review of "On Entropy Regularized Path Integral Control for Trajectory Optimization"

_entropy, 2020, doi:10.3390/e22101120_

Round 1
Reviewer 1 Report
This paper presents a series of theoretical results with the aim to provide novel insights and connections between several research lines, including evolutionary search, optimization, and control/reinforcement learning. The authors also propose an algorithm inspired in path integral (PI) control of potential practical interest, but without numerical experiments.
Overall, the paper is clearly written, although some parts require additional background in optimal control or reinforcement learning. The paper could also benefit from a more unifying notation, for example, considering the static formulation as the single-step dynamic formulation with end cost only. This will largely improve the readability and could make the connection between both formulations more explicit.
The main concerns with this paper are the lack of novelty and significance of the presented results. This would be a minor issue had the proposed algorithm been implemented and evaluated experimentally in some interesting domain(s). There are also some technical issues that need clarification.
Regarding the first point, the use of entropic regularization is quite common in (online convex) optimization, e.g., mirror descent algorithm. While this work focuses in gradient-free optimization, the proof techniques and the analysis leading to the proposed algorithms make extensive use of variational calculus, so one wonders why not already use the existing theory from the former formulations. From a control/RL perspective, the significance of the theoretical results looks rather limited, given the recent progress in the formulation of control as inference. The proposed algorithms optimize expected reward with trust regions and additional entropy constraints, a setting of widespread use nowadays. One of the main open questions here is how to improve sample efficiency, and is it unclear how the results in this paper contribute to that. Closely related works are MPO [1] or the work in [2]. The latter uses the same constraints as the ones proposed in this paper.
In light of the above paragraphs, while I think this work could have potential interest to the evolutionary search community, its value for the optimization and control/RL communities stands rather limited. I therefore think necessary to reframe the scope and narrative of the paper.
A second major concern is that authors disregard important advances and generalizations of PI control beyond PI^2 variants. Important examples are the work of Theodorou's group [7] which enables an adaptive update of control cost weight based on explicit uncertainty of the learned dynamics, and the extension to state-dependent feedback control from Kappen's group [4]. But most related to this work is the cross-entropy method [3] which extends the applicability of PI control to several control domains [5,6] and has been further extended using trust regions [9]. The authors should state explicitly in what way the attempted generalizations of PI control actually generalize the existing results for this class of SOC problems. Additionally, the assumptions should be made more explicit, in particular, when controls are not constrained to act in the same subspace as the noise, and when the existence of a known dynamics model is assumed and when an inverse dynamic function is sufficient. Given that there are no experiments, a more in depth qualitative comparison with [7], or at least [8], seems necessary.
Other remarks:
- Pag.3: "the observe" -> "then observe".
- The inverse temperature lambda is ignored in Theorem 1. How does the result change for a finite temperature? I suspect the optimum is not a Dirac delta in that case.
- Unify notation: the paper uses r, r_t or C for immediate cost and L, R for accumulated state-dependent cost.
- Eq. (9) looks like a general optimization problem but depends on \pi_g (previous iterate). This should be stated more clearly.
- Line 107: "is that what" -> "is the one that"
- Line 255: Define EOC.
- Line 341: "is essentially a conventional (..) problems" -> "is essentially a conventional (..) problem".
- Line 231: "optimal temporal stochastic optimal policy" -> "optimal time-dependent stochastic policy".
- Line 353: "A similar procedure has been proposed by Williams [28] and Drews [46] and can be considered state-of-art in PIC". (see references above).
- Lines 437-441 need rephrasing: "Levine et al. nor Neu et al. consider any iterative procedures": all algorithms in these papers are iterative.
Also, "Neither do ...": Levine's work draws the connection with Ziebart's quite explicitly and Neu shows that REPS is Mirror Descent with the relative entropy.
Refs:
[1] Model-Free Trajectory-based Policy Optimization with Monotonic Improvement. Akrour et al. JMLR (2018)
[2] Maximum a Posteriori Policy Optimisation. Abdolmaleki et al. ICLR (2018)
[3] Adaptive importance sampling for control and inference. Kappen and Ruiz. JSTAT (2016)
[4] Path integral control and state-dependent feedback. Thijssen and Kappen. PRE (2015)
[5] Particle smoothing for hidden diffusion processes: Adaptive path integral smoother. Ruiz and Kappen IEEE Trans Sig Proc (2017)
[6] Action selection in growing state spaces: control of network structure growth. Thalmeier et al. JPHYSA (2017)
[7] Sample Efficient Path Integral Control under Uncertainty. Pan et al. NeurIPS (2015)
[8] Policy search for path integral control. Gomez et al. ECML (2014)
[9] Adaptive Smoothing Path Integral Control. Thalmeier et al (2019)
[10] Policy search for motor primitives in robotics. Kober and Peters. NeurIPS (2009)
Reviewer 2 Report
Strength
- Broad literature survey and intuitive mathematical derivations throughout the paper help the readers grasp the big picture of the theory around the linearly solvable MDPs and path integral control.
- The entropic trajectory optimization algorithm presented in Sections 5.2.2 and 5.2.3 is novel to the best of my knowledge.
Weakness
- A large portion of the manuscript simply discusses known results. This is fine as long as one of the purposes of this paper is to give a holistic view of related subjects. However, the way it is written makes it hard to identify the authors' novel contributions. It would be very helpful if the authors could itemize their contributions in the abstract and the introduction section. They are somewhat summarized in the last three paragraphs in the conclusion section. Unfortunately, they are long and vague, and I'm not sure if they are really the authors' contributions. Also, some of the math results (Lemma 1, Lemma 2, Theorem 3, etc.) are standard. The authors should clearly distinguish the authors' new results from the existing results.
- There are many statements/sentences that are subjective (authors' personal opinions), rather than based on scientific facts or accurate literature survey. See comments 1, 2, 3 and 4 below.
Comments
1. Line 6 "So far Path Integral Control methods have been derived solely from the theory of Linearly Solvable Optimal Control.": Although the authors repetitively make this claim throughout the paper (line 80, line 320, etc), I do not understand what the authors try to mean by this. If I take it literally, I would disagree since ref 18 predates ref 16. Not being the author of ref 18, I do not know how the Path Integral Control method was derived. However, I believe it is possible to come up with the method by invoking the Feynman-Kac formula together with the fact that the HJB PDE is linearizable by the Cole-Hopf transformation.
2. Line 107 "..., to date the concept of information is not well defined." This sentence is misleading (as it may sound to denounce the whole Information Theory), and the paragraph to follow does not make the definition clearer.
3. Line 39 "Broadly speaking, ...": I’m not sure if this claim is accurate.
4. Line 259 "This particular subset ...": I’m not sure if this claim is accurate.
5. There are a few places in which the authors must either assume the space \mathcal{X} is finite, or include more rigorous discussions:
- Theorem 1: \pi converges to \delta in what sense? I do not think the authors should ever use the \lim notation without specifying the notion of convergence.
- Problem (9): What is the uniform distribution in R^n? Is \mathcal{U} the Lebesgue measure in this case? Also, how should one pick \epsilon? I do not think (9) yields any sensible solution if \epsilon is "too small."
6. Line 302: The existence of the inverse h_t is a strong assumption. Both the LQG setting of Kappen 2005 and the linearly solvable MDP of reference 16 are exceptional cases in which this inverse exists. Since this assumption is rarely met in practice, how to relax this assumption without destroying the linearly-solvable mature of the problem seems to be an important direction to investigate.
7. Line 298 "we formulate the general optimal control problem as a state trajectory optimization problem and consider the inputs to be implicit.": This indeed simplifies the problem and requires the existence of the inverse h_t. Alternatively, there are methods for entropic optimal control that do not make the control policy implicit:
[a] Rubin et al. "Trading value and information in MDPs" 2012.
[b] Tanaka et al. "Transfer-Entropy-Regularized Markov Decision Processes" arXiv 1708.09096.
[c] Lasry and Lions, "Mean field games" 2007 (and other methods for mean-field games where forward-backward algorithms are used).
8. Section 4.2: I had hard time understanding this entire section. What do you mean by "it is impossible" (line 242)? I did not understand the role of the "naive optimal path probability," which the authors claim to be different from the exact solution (Theorem 2) to the problem (13).
9. Line 341 "Since (15) is essentially ... a deterministic policy": I do not understand why (15) necessarily admits a deterministic policy. The linearly-solvable MDP in ref 16 admits the Boltzmann distribution as the optimal policy, which is not deterministic.
10. Sections 5.2.2 and 5.2.3: I assume the contents of these sections are (one of) the main contributions of this paper. If this is the case, more comprehensive derivations/explanations should be included.
Typos:
1. Line 107: "that what" -> "what"
2. Line 480: "infor-mation" -> "information"
